# Association between minerals intake and childhood obesity: A cross-sectional study of the NHANES database in 2007–2014

Lu Wang ⓘ*, Wei Liu, Sitong Bi, Li Zhou, Lihua Li

Department of pediatrics, Beijing Luhe Hospital Affiliated to Capital Medical University, Beijing, P.R. China

* luluwangbj@outlook.com

## Abstract

### Background

The roles of minerals in obesity received increasing attention recently due to its oxidant or antioxidant functions and effects on insulin and glucose metabolism that may be associated with obesity. Herein, this study aims to explore the association between minerals and obesity and body mass index (BMI) in children with different ages, and hope to provide some references for prevention and management in children with high-risk of obesity.

### Methods

Data of children aged 2–17 years old were extracted from the National Health and Nutrition Examination Survey (NHANES) database in 2007–2014 in this cross-sectional study. Weighted univariate and multivariate logistic regression and liner regression analyses were used to screen covariates, and explore the association between minerals [including calcium (Ca), phosphorus (P), magnesium (Mg), iron (Fe), zinc (Zn), copper (Cu), sodium (Na), potassium (K) and selenium (Se)] and childhood obesity and BMI. The evaluation indexes were β, odds ratios (ORs) and 95% confidence intervals (CIs). These relationships were also investigated in age subgroups.

### Results

Among 10,450 eligible children, 1,988 (19.02%) had obesity. After adjusting for covariates, we found the highest quartile of dietary Fe [OR = 0.74, 95%CI: (0.58, 0.95)] and Zn [OR = 0.70, 95%CI: (0.54, 0.92)] intakes were associated with low odds of childhood obesity, while that of dietary Na intake seemed to be positively linked to childhood obesity [OR = 1.35, 95%CI: (1.05, 1.74)]. High dietary intakes of Ca, Na and K were positively associated with children's BMI, on the contrary, dietary Fe and Zn consumptions had a negative one (all $P<0.05$). Additionally, these associations were also found in children with different age (all $P<0.05$).

**Data Availability Statement:** The datasets used and/or analyzed during the current study are available from the NHANES database, https://www.cdc.gov/nchs/nhanes/index.htm.

**Funding:** The author(s) received no specific funding for this work.

**Competing interests:** The authors have declared that no competing interests exist.

## Conclusion

Dietary Fe and Zn intakes played positive roles in reducing childhood obesity or BMI, while the intakes of Na should be controlled suitably.

## Introduction

The global prevalence of childhood obesity has grown sharply in recent decades [1], and generated an enormous individual and socioeconomic burden [2,3]. Approximately 70 million children will be obese or overweight in developing countries by 2025 [4]. Obesity in children has contributed to the increased risk of chronic diseases, such as obesity in adulthood, mental health problems, diabetes mellitus (DM), cardiovascular disease (CVD), some types of cancer, and death [5]. Therefore, the prevention and treatment strategies on childhood obesity need to be given primary importance.

Dietary/nutritional intervention plays a central role in the prevention of obesity [6]. In the past, most dietary measures on weight control focused on reducing the intake of macronutrients such as carbohydrates and fats [7,8]. Recently, the effects of minerals on obesity have received increasing attention [9]. Minerals can alter the composition of the intestinal microbiota, gut barrier function, compartmentalized metabolic inflammation, cellular glucose transport, and endocrine control of glucose metabolism, which may be associated with the occurrence and development of obesity [10,11]. Transition metals such as iron (Fe), zinc (Zn), copper (Cu) and selenium (Se) play important roles in cell metabolism, and their oxidant/antioxidant functions may be involved in the mechanism of obesity [12]. Also, sodium (Na) as a macroelement has been reported to be positively associated with the risk of obesity [13], which may affect insulin and glucose metabolism, accelerate leptin production or secretion and enhance leptin resistance, leading to energy imbalance, accumulation of adipose tissue mass and eventually obesity [14]. Gu et al. [15,16] found that serum Zn levels in children with obesity were significantly lower than those without obesity, while the serum Cu levels were higher. However, studies extensively discussing the association between dietary minerals intake and the risk of childhood obesity are still absent.

Herein, this study based on the National Health and Nutrition Examination Survey (NHANES) database, with the aim of exploring association between nine common minerals including calcium (Ca), phosphorus (P), magnesium (Mg), Fe, Zn, Cu, Na, potassium (K) and Se, and obesity and body mass index (BMI) in children, in order to provide some dietary references for prevention and management of childhood obesity.

## Methods

### Study design and population

Data of children in this cross-sectional study were extracted from the NHANES database in 2007–2014. The NHANES is a multipurpose research program done by the National Center for Health Statistics (NCHS) to assess the health and nutritional status of population in the United States [17]. It collects data of approximately 5,000 persons from 15 areas regularly since 1999 that includes a household interview followed by a standardized physical examination in a mobile examination center (MEC). A stratified multistage sampling design with a weighting scheme based on the selection of counties, blocks, households, and persons within households is used by NHANES to represent the civilian, non-institutionalized population in the United

States and accurately estimate disease prevalence (https://www.cdc.gov/nchs/nhanes/index.htm).

We initially included 13,058 children aged 2–17 years old from the database. Then, those who without information of minerals intake, BMI, or poverty income ratio (PIR) were excluded. Finally, 10,450 children were eligible. The NHANES survey was approved by the institutional review board (IRB) of NCHS. The participants' legal guardians/next of kin have provided written informed consent for participation. Since all the data were de-identified and publicly available, no ethical approval from the IRB of Beijing Luhe Hospital Affiliated to Capital Medical University was required.

## Assessment of dietary intake of minerals

NHANES collects the dietary status of participants through two 24-hour dietary recall interviews, which is according to the United States Department of Agriculture (USDA) automated multiple-pass method (AMPM) [18]. The first dietary recall interview was conducted in person, and the second one was conducted 3–10 days later via a phone call. During the interviews, consumption frequency, duration, and dosage were recorded for each of the dietary, dietary supplements, and prescription medication over the prior 30 days using questionnaires, which can be used for calculation of the average daily intake of nutrients. Respondents for the dietary interviews included the following: a proxy for child aged <6 years old; a proxy with the assistance of the child for those aged from 6 to 8 years old; assistance of a proxy for child aged 9–11 years old; and children aged ≥12 years old who answered by themselves.

In the current study, we extracted the data on dietary mineral intake and its supplements in the first 24-hour recall, and standardized by the total energy intake: mineral intake/total energy (mg/1000kcal). The dietary Ca (mg), P (mg), Mg (mg), Fe (mg), Zn (mg), Cu (mg), Na (mg), K (mg) and Se (mcg) intakes were divided into four levels according to their respective quartiles.

## Obesity diagnosis and BMI measurement

The children BMI was calculated using the formula: weight/height$^2$ (kg/m$^2$). Obesity was judged by the BMI z-score, the CDC recommended percentiles, which was calculated accounting for age and sex. A BMI z-score ≥95th percentile indicates obesity. For more details of the calculation of the BMI z-score please visit the NHANES website: https://www.cdc.gov/healthyweight/assessing/bmi/childrens_bmi/about_childrens_bmi.html.

## Covariates

We also collected variables including age, gender, race, PIR, physical activity, sedentary time, maternal smoking during pregnancy, cotinine (ng/mL), and the intakes of protein (gm/1000kcal), carbohydrate (gm/1000kcal) and fat (gm/1000kcal) from the database. Physical activity of children aged 12–17 years old was translated into energy expenditure. The Metabolic equivalent (MET) was calculated based on information collected from the NHANES questionnaire of physical activity reports (PAQ) [19]: Energy expenditure (MET·min) = recommended MET × exercise time of corresponding activity (min). The ideal physical activity was ≥180 MET·min/day (for children aged 12–17 years old or ≥60 min/day (for children aged 2–11 years old), and otherwise recognized as not achieve the ideal physical activity. Sedentary time (time watching TV or video or using a computer) per average day over the last 30 days was collected through the household interview. The sedentary time was divided into four levels: <3 hours, 3–6 hours, ≥6 hours and unknown. The detection limit of cotinine is 0.011 ng/mL. The intakes of total energy, carbohydrate, protein and fat were collected according to both food and supplements respectively from the first 24-hour dietary recall in the NHANES.

## Statistical analysis

The continuous variables were described using mean ± standard error (mean ± SE), and t test was used for comparison between groups. Categorical data were expressed using frequency and constituent ratio [N (%)], and chi-square test ($\chi^2$) was used for the comparison. A set of NHANES special weights "WTDRD1" were used for the analyses because we included data on the first 24-hour dietary recall in dietary minerals intake assessment. These weights were constructed by taking the 2-year cycle MEC sample weights (WTMEC2YR) and further adjusting for (a) the additional non-response and (b) the differential allocation by day of the week for the dietary intake data collection.

When explored the association between dietary minerals intake and childhood obesity, weighted univariate and multivariate logistic regression analyses were used. Adjusted model adjusted covariates including age, gender, race, PIR, physical activity, sedentary time, maternal smoking during pregnancy, cotinine, carbohydrate intake, protein intake and fat intake. The evaluation index was odds ratios (ORs) and 95% confidence intervals (CIs). When explored the association between dietary minerals intake and BMI weighted univariate and multivariate liner regression analyses were performed. Adjusted model adjusted covariates including age, gender, race, PIR, physical activity, sedentary time, and maternal smoking during pregnancy. The evaluation index was the estimated value (β) and 95% confidence intervals (CIs). We also explored these relationships in subgroups of different ages (1–5, 6–11, and 12–17 years old). The reference group in the dependent variable was the lowest quartile of each dietary mineral intake level.

Two-sided $P<0.05$ was considered significant. Statistical analysis was performed using SAS 9.4 (SAS Institute, Cary, NC, USA). Missing variables, including physical activity, sedentary time, maternal smoking during pregnancy, and cotinine, were classified into the "unknown" category.

## Results

### Characteristics of children

Fig 1 is the flowchart of participants screening. There were 13,058 children aged 2–17 years old in the NHANES database in 2007–2014. Then, children without the information of minerals intake (n = 1661), BMI (n = 138), or PIR (n = 809) were excluded. Finally, 10,450 were eligible.

The characteristics of eligible children are showed in Table 1. Among the eligible children, 1,988 (19.02%) had obesity. The average age of total children was 9.56 years old. More than half of children with obesity were male [1048 (53.49%)], while those without obesity included a higher percentage of females [4150 (50.12%)]. In children without obesity, 4,126 (44.74%) had an ideal physical activity level, while this number in children with obesity was 741 (31.90%). The dietary intake levels of Ca (578.39 vs. 559.39 mg/1000kcal), Fe (8.30 vs. 7.65 mg/1000kcal), Zn (6.15 vs. 5.79 mg/1000kcal), Cu (0.62 vs. 0.58 mg/1000kcal), Na (1572.30 vs. 1645.84 mg/1000kcal), and Se (49.35 vs. 50.92 mcg/1000kcal) were significantly different between non-obesity group and obesity group. In addition, race, PIR, sedentary time, maternal smoking during pregnancy, cotinine, and intakes of protein and carbohydrate were also significantly different between non-obesity group and obesity group (all $P<0.05$).

### Association between dietary minerals intake and childhood obesity

We explored the association between dietary minerals intake and childhood obesity (Table 2). After adjusting for covariates, we found that compared with the lowest quartiles, the highest

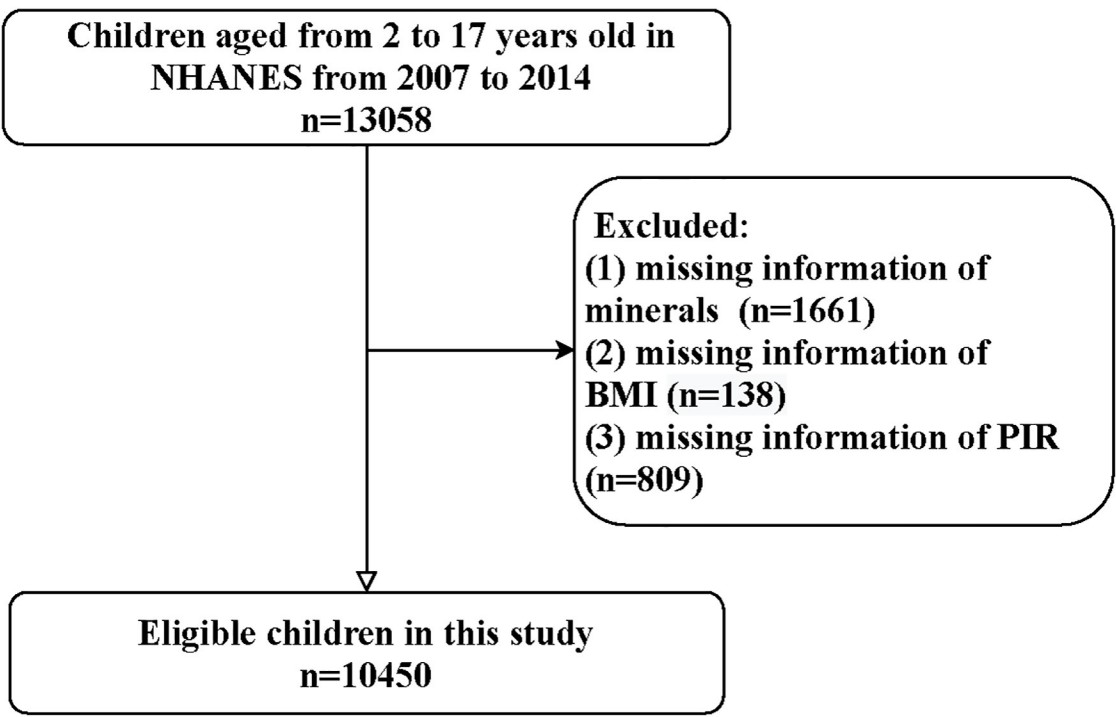

**Fig 1. Flowchart of the participants screening.**

quartile of dietary Fe [OR = 0.74, 95% CI: (0.58, 0.95)] and Zn [OR = 0.70, 95% CI: (0.54, 0.92)] intakes were associated with lower odds of childhood obesity, while the highest quartile of dietary Na intake [OR = 1.35, 95% CI: (1.05, 1.74)] was associated with higher odds of childhood obesity.

## Association between dietary minerals intake and childhood obesity in age subgroups

Table 3 shows the association between dietary minerals intake and childhood obesity in age subgroups. The highest quartile of dietary intake of Zn was associated with lower odds of childhood obesity in children aged 6–11 years old [OR = 0.54, 95% CI: (0.36, 0.80)], compared with the lowest quartile. Differently, higher level of Mg [OR = 1.56, 95% CI: (1.05, 2.32)] and Na [OR = 1.58, 95% CI: (1.11, 2.26)] consumptions were associated with higher odds of childhood obesity. In children aged 2–5 years old or 12–17 years old, these relationships were not significantly.

## Association between dietary minerals intake and BMI

We also explored the association between dietary minerals intake and children's BMI (Table 4). After adjusting for covariates, dietary intake of Ca [β = 0.50, 95% CI: (0.07, 0.94)], Na [β = 0.48, 95% CI: (0.04, 0.91)] and K [β = 0.62, 95% CI: (0.19, 1.05)] were all positively associated with BMI, while Fe [β = -0.78, 95% CI: (-1.17, -0.39)] and Zn [β = -0.56, 95% CI: (-1.01, -0.11)] had negative associations with BMI.

**Table 1. Characteristics of eligible children.**

| Variables | Total (n = 10450) | Non-obesity (n = 8462) | Obesity (n = 1988) | Statistics | P |
|---|---|---|---|---|---|
| Age, years, Mean (S.E) | 9.56 (0.08) | 9.33 (0.08) | 10.59 (0.14) | t = -8.86 | <0.001 |
| Age levels, n (%) | | | | $\chi^2$ = 85.13 | <0.001 |
| 2–5 | 2888 (24.25) | 2564 (26.72) | 324 (12.89) | | |
| 6–11 | 4172 (37.68) | 3272 (36.58) | 900 (42.73) | | |
| 12–17 | 3390 (38.07) | 2626 (36.70) | 764 (44.38) | | |
| Gender, n (%) | | | | $\chi^2$ = 3.99 | 0.046 |
| Male | 5360 (50.52) | 4312 (49.88) | 1048 (53.49) | | |
| Female | 5090 (49.48) | 4150 (50.12) | 940 (46.51) | | |
| Race, n (%) | | | | $\chi^2$ = 74.87 | <0.001 |
| Mexican American | 2427 (14.25) | 1880 (12.98) | 547 (20.11) | | |
| Other Hispanic | 1171 (7.44) | 908 (7.03) | 263 (9.33) | | |
| Non-Hispanic White | 3099 (56.57) | 2612 (58.38) | 487 (48.23) | | |
| Non-Hispanic Black | 2600 (13.96) | 2069 (13.58) | 531 (15.72) | | |
| Other Race—including multi-racial | 1153 (7.77) | 993 (8.03) | 160 (6.62) | | |
| PIR, Mean (S.E) | 2.44 (0.07) | 2.51 (0.07) | 2.13 (0.08) | t = 4.57 | <0.001 |
| Physical activity, n (%) | | | | $\chi^2$ = 49.47 | <0.001 |
| Ideal physical activity | 4867 (42.45) | 4126 (44.74) | 741 (31.90) | | |
| Not ideal physical activity | 2570 (24.14) | 1994 (22.97) | 576 (29.52) | | |
| Unknown | 3013 (33.40) | 2342 (32.28) | 671 (38.58) | | |
| Sedentary time levels, n (%) | | | | $\chi^2$ = 61.84 | <0.001 |
| <3 | 1915 (17.08) | 1654 (18.56) | 261 (10.23) | | |
| 3–6 | 1227 (10.27) | 984 (10.23) | 243 (10.44) | | |
| ≥6 | 1955 (20.96) | 1485 (19.47) | 470 (27.80) | | |
| Unknown | 5353 (51.70) | 4339 (51.73) | 1014 (51.53) | | |
| Maternal smoking during pregnancy, n (%) | | | | $\chi^2$ = 10.42 | 0.005 |
| No | 8057 (73.94) | 6574 (74.98) | 1483 (69.13) | | |
| Yes | 1160 (11.99) | 909 (11.37) | 251 (14.89) | | |
| Unknown | 1233 (14.07) | 979 (13.65) | 254 (15.98) | | |
| Cotinine, ng/mL, Mean (S.E) | 4.53 (0.65) | 4.55 (0.66) | 4.44 (1.50) | t = 0.08 | 0.938 |
| Cotinine levels, n (%) | | | | $\chi^2$ = 40.02 | <0.001 |
| 0.011 | 2047 (22.37) | 1667 (22.84) | 380 (20.21) | | |
| >0.011 | 5696 (54.24) | 4423 (52.04) | 1273 (64.39) | | |
| Unknown | 2707 (23.38) | 2372 (25.11) | 335 (15.40) | | |
| Ca, mg/1000kcal, Mean (S.E) | 575.01 (5.13) | 578.39 (5.31) | 559.39 (7.98) | t = 2.47 | 0.016 |
| P, mg/1000kcal, Mean (S.E) | 677.59 (3.46) | 678.24 (3.70) | 674.61 (5.98) | t = 0.58 | 0.564 |
| Mg, mg/1000kcal, Mean (S.E) | 127.24 (0.73) | 127.14 (0.76) | 127.66 (1.67) | t = -0.30 | 0.767 |
| Fe, mg/1000kcal, Mean (S.E) | 8.18 (0.09) | 8.30 (0.10) | 7.65 (0.13) | t = 4.49 | <0.001 |
| Zn, mg/1000kcal, Mean (S.E) | 6.09 (0.08) | 6.15 (0.09) | 5.79 (0.12) | t = 2.50 | 0.015 |
| Cu, mg/1000kcal, Mean (S.E) | 0.61 (0.01) | 0.62 (0.01) | 0.58 (0.01) | t = 2.57 | 0.013 |
| Na, mg/1000kcal, Mean (S.E) | 1585.40 (8.37) | 1572.30 (9.77) | 1645.84 (17.90) | t = -3.45 | <0.001 |
| K, mg/1000kcal, Mean (S.E) | 1188.07 (6.81) | 1185.58 (7.20) | 1199.54 (15.65) | t = -0.85 | 0.398 |
| Se, mcg/1000kcal, Mean (S.E) | 49.63 (0.29) | 49.35 (0.34) | 50.92 (0.48) | t = -2.64 | 0.010 |
| Protein, gm/1000kcal, Mean (S.E) | 35.90 (0.19) | 35.67 (0.22) | 36.96 (0.29) | t = -3.77 | <0.001 |
| Carbohydrate, gm/1000kcal, Mean (S.E) | 135.73 (0.44) | 136.13 (0.48) | 133.85 (0.87) | t = 2.37 | 0.021 |

*(Continued)*

**Table 1.** (Continued)

| Variables | Total (n = 10450) | Non-obesity (n = 8462) | Obesity (n = 1988) | Statistics | P |
|---|---|---|---|---|---|
| Fat, gm/1000kcal Mean (S.E) | 36.16 (0.15) | 36.10 (0.17) | 36.46 (0.31) | t = -1.07 | 0.291 |

t: Test, $\chi^2$: Chi-square test.

S.E: Standard error, PIR: Poverty-income ratio, Ca: Calcium, P: Phosphorus, Mg: Magnesium, Fe: Iron, Zn: Zinc, Cu: Copper, Na: Sodium, K: Potassium and Se: Selenium.

### Association between dietary minerals intake and BMI in age subgroups

These associations were further explored in age subgroups, and the results were showed in Table 5. In children aged 2–5 years old, we only found a negative association between dietary Fe intake and BMI [β = -0.29, 95% CI: (-0.56, -0.03)]. In those who aged 6–11 years old, dietary intake of Zn was negatively linked to BMI [β = -1.04, 95% CI: (-1.65, -0.43)], while dietary intake of Na was positively linked to BMI [β = 0.65, 95% CI: (0.05, 1.25)]. In the 12–17 years old group, dietary Fe intake was negatively associated with BMI [β = -1.31, 95% CI: (-2.11, -0.52)], and similarly to those who aged 6–11 years old, relationship between dietary Na intake and BMI was positive [β = 0.84, 95% CI: (0.01, 1.66)].

## Discussion

This study explored the association between dietary intakes of nine common minerals and obesity and BMI in children. The results showed that higher level of dietary Fe and Zn intakes were associated with lower odds of childhood obesity. Oppositely, higher levels of dietary Cu and Na intakes seemed to associated with higher odds of obesity. Dietary intakes of Ca, Na and K were positively linked to the children's BMI, whereas dietary Fe and Zn consumptions shared negatively associations with BMI. Additionally, these relationships were also found in children with different age.

Findings in the current study supported hypotheses that dietary intake of Fe and Zn levels may be potential protective factors in the development of childhood obesity, whereas Cu, Na, Ca, Mg and K played opposite roles. Fan et al. [20] explored the relationship between serum metallic elements and obesity in children in the United States, and found the highest quartile of Cu concentrations in blood with an obesity status, while a negative association existed between blood Zn and obesity. We came to the same conclusion with Fan et al. in this study, and however, we used the dietary intake of metallic elements other than the serum concentrations, and were standardized by the total energy intake. A meta-analysis suggested that children with obesity to have significantly different markers of Fe deficiency than those in control group [21]. Specially, children with obesity had significantly lower Fe, transferrin saturation, and total-iron binding capacity along with higher ferritin, soluble transferrin receptors and hepcidin-25 than children of normal weight [21]. In this study, we similarly discovered higher level of dietary Fe intake was associated with lower odds of childhood obesity, but the underlying mechanism is needed further exploration. A cross-sectional study based on the NHANES database by Zhao et al. [22] showed that the highest quartile of Na intake was positively associated with overweight, obesity, and central obesity among children in the United States. In the same way, results of the current study found the relationships between higher dietary intake level of Na and childhood obesity as well as higher BMI in children. Besides, Wang et al. [23] discovered whole-blood Mg concentration was an independent risk factor of overweight or

**Table 2. Association between dietary minerals intake and childhood obesity.**

| Dietary minerals intake | Children with obesity/total samples | Crude model OR (95% CI) | Adjusted model# OR (95% CI) |
|---|---|---|---|
| Ca (mg/1000kcal) | | | |
| ≤391.56 | 580 / 2939 | Ref | Ref |
| 391.56–544.69 | 521 / 2687 | 0.94 (0.76–1.16) | 0.98 (0.78–1.23) |
| 544.69–721.81 | 485 / 2473 | 0.95 (0.78–1.17) | 1.04 (0.85–1.28) |
| >721.81 | 402 / 2351 | 0.82 (0.68–1.00) | 0.99 (0.79–1.25) |
| P (mg/1000kcal) | | | |
| ≤549.25 | 578 / 2829 | Ref | Ref |
| 549.25–661.63 | 502 / 2777 | 0.81 (0.68–0.96) * | 0.76 (0.63–0.92) ** |
| 661.63–786.65 | 500 / 2545 | 1.03 (0.85–1.25) | 1.02 (0.82–1.28) |
| >786.65 | 408 / 2299 | 0.92 (0.78–1.09) | 0.92 (0.70–1.21) |
| Mg (mg/1000kcal) | | | |
| ≤100.41 | 555 / 2770 | Ref | Ref |
| 100.41–122.18 | 494 / 2673 | 0.98 (0.78–1.23) | 1.04 (0.81–1.33) |
| 122.18–146.32 | 475 / 2547 | 1.03 (0.84–1.28) | 1.13 (0.90–1.42) |
| >146.32 | 464 / 2460 | 0.97 (0.77–1.22) | 1.09 (0.85–1.39) |
| Fe (mg/1000kcal) | | | |
| ≤5.39 | 554 / 2618 | Ref | Ref |
| 5.39–6.85 | 495 / 2562 | 0.98 (0.81–1.18) | 0.97 (0.80–1.19) |
| 6.85–9.33 | 508 / 2637 | 0.91 (0.75–1.11) | 0.88 (0.72–1.09) |
| >9.33 | 431 / 2633 | 0.73 (0.57–0.92) ** | 0.74 (0.58–0.95) * |
| Zn (mg/1000kcal) | | | |
| ≤4.01 | 560 / 2771 | Ref | Ref |
| 4.01–5.20 | 514 / 2646 | 0.88 (0.74–1.03) | 0.85 (0.72–1.01) |
| 5.20–7.08 | 490 / 2551 | 0.91 (0.76–1.10) | 0.86 (0.67–1.11) |
| >7.08 | 424 / 2482 | 0.75 (0.61–0.94) * | 0.70 (0.54–0.92) * |
| Cu (mg/1000kcal) | | | |
| ≤0.41 | 524 / 2805 | Ref | Ref |
| 0.41–0.50 | 523 / 2558 | 1.23 (0.98–1.54) | 1.28 (1.02–1.61) * |
| 0.50–0.64 | 548 / 2760 | 1.20 (0.93–1.54) | 1.29 (1.01–1.66) * |
| >0.64 | 393 / 2327 | 0.89 (0.69–1.15) | 1.04 (0.80–1.37) |
| Na (mg/1000kcal) | | | |
| ≤1282.86 | 460 / 2664 | Ref | Ref |
| 1282.86–1541.84 | 489 / 2703 | 1.22 (0.95–1.57) | 1.16 (0.90–1.50) |
| 1541.84–1819.50 | 462 / 2564 | 1.13 (0.91–1.40) | 1.01 (0.80–1.27) |
| >1819.50 | 577 / 2519 | 1.60 (1.28–2.01) *** | 1.35 (1.05–1.74) * |
| K (mg/1000kcal) | | | |
| ≤916.70 | 526 / 2693 | Ref | Ref |
| 916.70–1141.73 | 533 / 2662 | 1.15 (0.93–1.42) | 1.17 (0.94–1.46) |
| 1141.73–1407.27 | 460 / 2554 | 1.07 (0.86–1.34) | 1.20 (0.95–1.53) |
| >1407.27 | 469 / 2541 | 1.10 (0.89–1.35) | 1.27 (1.00–1.61) |
| Se (mcg/1000kcal) | | | |
| ≤37.75 | 463 / 2580 | Ref | Ref |
| 37.75–48.05 | 484 / 2773 | 1.08 (0.84–1.38) | 1.01 (0.77–1.33) |
| 48.05–58.52 | 508 / 2501 | 1.23 (0.96–1.58) | 1.10 (0.82–1.47) |

(*Continued*)

**Table 2.** (Continued)

| Dietary minerals intake | Children with obesity/total samples | Crude model OR (95% CI) | Adjusted model# OR (95% CI) |
|---|---|---|---|
| >58.52 | 533 / 2596 | 1.30 (1.04–1.62) * | 1.05 (0.78–1.40) |

OR: Odd ratio, CI: Confidence interval, Ca: Calcium, Ref: Reference, P: Phosphorus, Mg: Magnesium, Fe: Iron, Zn: Zinc, Cu: Copper, Na: Sodium, K: Potassium and Se: Selenium

*P<0.05,

**P<0.01,

***P<0.001.

# Adjusted for age, gender, race, PIR, physical activity, sedentary time, maternal smoking during pregnancy, cotinine, carbohydrate intake, protein intake and fat intake.

Note: We used g/1000kcal as the units of Ca, P, Mg, Na and K to expanded the value.

**Table 3.  Association between dietary minerals intake and childhood obesity in age subgroups.**

| Dietary minerals intake | 2–5 years old OR (95% CI) | 6–11 years old OR (95% CI) | 12–17 years old OR (95% CI) |
|---|---|---|---|
| Ca (mg/1000kcal) | | | |
| ≤391.56 | Ref | Ref | Ref |
| 391.56–544.69 | 0.94 (0.56–1.58) | 0.98 (0.72–1.33) | 0.95 (0.67–1.34) |
| 544.69–721.81 | 0.78 (0.44–1.36) | 1.03 (0.78–1.36) | 1.09 (0.79–1.51) |
| >721.81 | 0.80 (0.45–1.42) | 0.95 (0.63–1.43) | 1.11 (0.74–1.65) |
| P (mg/1000kcal) | | | |
| ≤549.25 | Ref | Ref | Ref |
| 549.25–661.63 | 0.72 (0.39–1.33) | 0.89 (0.68–1.16) | 0.64 (0.44–0.92) * |
| 661.63–786.65 | 0.72 (0.41–1.25) | 0.99 (0.73–1.33) | 1.14 (0.81–1.63) |
| >786.65 | 0.68 (0.35–1.32) | 0.88 (0.60–1.30) | 1.01 (0.65–1.57) |
| Mg (mg/1000kcal) | | | |
| ≤100.41 | Ref | Ref | Ref |
| 100.41–122.18 | 0.93 (0.58–1.51) | 1.11 (0.83–1.49) | 1.08 (0.70–1.67) |
| 122.18–146.32 | 0.87 (0.52–1.47) | 1.11 (0.76–1.60) | 1.43 (1.00–2.04) |
| >146.32 | 0.93 (0.54–1.60) | 1.56 (1.05–2.32) * | 0.98 (0.66–1.45) |
| Fe (mg/1000kcal) | | | |
| ≤5.39 | Ref | Ref | Ref |
| 5.39–6.85 | 0.72 (0.45–1.14) | 1.35 (0.98–1.87) | 0.78 (0.55–1.11) |
| 6.85–9.33 | 0.61 (0.40–0.93) * | 1.18 (0.86–1.62) | 0.75 (0.55–1.01) |
| >9.33 | 0.62 (0.38–1.00) | 0.97 (0.66–1.43) | 0.66 (0.43–1.00) |
| Zn (mg/1000kcal) | | | |
| ≤4.01 | Ref | Ref | Ref |
| 4.01–5.20 | 1.03 (0.68–1.55) | 0.69 (0.47–1.00) | 0.97 (0.67–1.42) |
| 5.20–7.08 | 1.14 (0.75–1.74) | 0.80 (0.57–1.13) | 0.91 (0.56–1.46) |
| >7.08 | 0.80 (0.49–1.30) | 0.54 (0.36–0.80) ** | 0.93 (0.61–1.42) |
| Cu (mg/1000kcal) | | | |
| ≤0.41 | Ref | Ref | Ref |
| 0.41–0.50 | 1.31 (0.84–2.04) | 1.49 (1.10–2.00) * | 1.05 (0.72–1.53) |
| 0.50–0.64 | 1.30 (0.87–1.94) | 1.62 (1.12–2.35) * | 1.06 (0.73–1.54) |
| >0.64 | 1.03 (0.60–1.76) | 1.29 (0.93–1.81) | 0.89 (0.57–1.38) |
| Na (mg/1000kcal) | | | |

(*Continued*)

**Table 3.** (Continued)

| Dietary minerals intake | 2–5 years old OR (95% CI) | 6–11 years old OR (95% CI) | 12–17 years old OR (95% CI) |
|---|---|---|---|
| ≤1282.86 | Ref | Ref | Ref |
| 1282.86–1541.84 | 1.26 (0.82–1.94) | 1.26 (0.89–1.77) | 1.08 (0.71–1.64) |
| 1541.84–1819.50 | 0.98 (0.58–1.66) | 1.04 (0.77–1.41) | 0.96 (0.67–1.36) |
| >1819.50 | 1.43 (0.87–2.33) | 1.58 (1.11–2.26) * | 1.20 (0.81–1.78) |
| K (mg/1000kcal) | | | |
| ≤916.70 | Ref | Ref | Ref |
| 916.70–1141.73 | 1.23 (0.74–2.05) | 1.17 (0.87–1.57) | 1.16 (0.80–1.70) |
| 1141.73–1407.27 | 0.84 (0.49–1.46) | 1.16 (0.81–1.65) | 1.46 (1.03–2.09) * |
| >1407.27 | 1.20 (0.75–1.92) | 1.44 (0.98–2.11) | 1.19 (0.81–1.76) |
| Se (mcg/1000kcal) | | | |
| ≤37.75 | Ref | Ref | Ref |
| 37.75–48.05 | 1.02 (0.61–1.68) | 0.98 (0.69–1.39) | 1.04 (0.66–1.65) |
| 48.05–58.52 | 0.94 (0.53–1.67) | 1.03 (0.72–1.48) | 1.18 (0.74–1.87) |
| >58.52 | 0.81 (0.47–1.40) | 1.02 (0.65–1.61) | 1.12 (0.71–1.75) |

OR: Odd ratio, CI: Confidence interval, Ca: Calcium, Ref: Reference, P: Phosphorus, Mg: Magnesium, Fe: Iron, Zn: Zinc, Cu: Copper, Na: Sodium, K: Potassium and Se: Selenium.

*$P<0.05$,

**$P<0.01$,

***$P<0.001$.

Adjusted for gender, race, PIR, physical activity, sedentary time, maternal smoking during pregnancy, cotinine, carbohydrate intake, protein intake and fat intake.

Note: We used g/1000kcal as the units of Ca, P, Mg, Na and K to expanded the value.

**Table 4. Association between dietary minerals intake and BMI in children.**

| Dietary minerals intake | Crude model β (95% CI) | Adjusted model[#] β (95% CI) |
|---|---|---|
| Ca (mg/1000kcal) | | |
| ≤391.56 | Ref | Ref |
| 391.56 to 544.69 | -0.49 (-0.91, -0.07) * | 0.10 (-0.25, 0.44) |
| 544.69 to 721.81 | -0.69 (-1.13, -0.25) ** | 0.24 (-0.15, 0.64) |
| >721.81 | -1.13 (-1.61, -0.65) *** | 0.50 (0.07, 0.94) * |
| P (mg/1000kcal) | | |
| ≤549.25 | Ref | Ref |
| 549.25 to 661.63 | -0.54 (-0.87, -0.20) ** | -0.36 (-0.72, 0.01) |
| 661.63 to 786.65 | -0.57 (-1.03, -0.10) * | 0.10 (-0.34, 0.53) |
| >786.65 | -0.59 (-1.01, -0.17) ** | 0.23 (-0.23, 0.68) |
| Mg (mg/1000kcal) | | |
| ≤100.41 | Ref | Ref |
| 100.41 to 122.18 | -0.65 (-1.12, -0.17) ** | -0.06 (-0.51, 0.40) |
| 122.18 to 146.32 | -0.71 (-1.22, -0.20) ** | 0.28 (-0.13, 0.69) |
| >146.32 | -0.41 (-0.90, 0.08) | 0.30 (-0.15, 0.76) |
| Fe (mg/1000kcal) | | |
| ≤5.39 | Ref | Ref |
| 5.39 to 6.85 | -0.28 (-0.70, 0.15) | -0.30 (-0.67, 0.08) |

*(Continued)*

**Table 4.** (Continued)

| Dietary minerals intake | Crude model β (95% CI) | Adjusted model# β (95% CI) |
|---|---|---|
| 6.85 to 9.33 | -0.78 (-1.18, -0.38) *** | -0.50 (-0.82, -0.19) ** |
| >9.33 | -1.28 (-1.72, -0.84) *** | -0.78 (-1.17, -0.39) *** |
| Zn (mg/1000kcal) | | |
| ≤4.01 | Ref | Ref |
| 4.01 to 5.20 | -0.59 (-1.05, -0.13) * | -0.16 (-0.50, 0.18) |
| 5.20 to 7.08 | -0.67 (-1.02, -0.32) *** | -0.21 (-0.64, 0.22) |
| >7.08 | -0.99 (-1.46, -0.53) *** | -0.56 (-1.01, -0.11) * |
| Cu (mg/1000kcal) | | |
| ≤0.41 | Ref | Ref |
| 0.41 to 0.50 | -0.02 (-0.57, 0.52) | 0.22 (-0.16, 0.60) |
| 0.50 to 0.64 | -0.11 (-0.65, 0.43) | 0.19 (-0.20, 0.58) |
| >0.64 | -0.52 (-1.04, 0.01) | -0.08 (-0.46, 0.31) |
| Na (mg/1000kcal) | | |
| ≤1282.86 | Ref | Ref |
| 1282.86 to 1541.84 | 0.47 (-0.03, 0.96) | 0.14 (-0.26, 0.53) |
| 1541.84 to 1819.50 | 0.59 (0.21, 0.97) ** | -0.15 (-0.46, 0.16) |
| >1819.50 | 2.01 (1.52, 2.51) *** | 0.48 (0.04, 0.91) * |
| K (mg/1000kcal) | | |
| ≤916.70 | Ref | Ref |
| 916.70 to 1141.73 | -0.43 (-0.79, -0.06) * | 0.14 (-0.22, 0.50) |
| 1141.73 to 1407.27 | -0.69 (-1.13, -0.25) ** | 0.40 (0.02, 0.77) * |
| >1407.27 | -0.89 (-1.33, -0.45) *** | 0.62 (0.19, 1.05) ** |
| Se (mcg/1000kcal) | | |
| ≤37.75 | Ref | Ref |
| 37.75 to 48.05 | -0.11 (-0.58, 0.35) | -0.21 (-0.65, 0.23) |
| 48.05 to 58.52 | 0.18 (-0.33, 0.68) | -0.34 (-0.85, 0.17) |
| >58.52 | 1.22 (0.70, 1.73) *** | -0.09 (-0.60, 0.42) |

BMI: Body mass index, CI: Confidence interval, Ca: Calcium, Ref: Reference, P: Phosphorus, Mg: Magnesium, Fe: Iron, Zn: Zinc, Cu: Copper, Na: Sodium, K: Potassium and Se: Selenium.

*$P<0.05$,

**$P<0.01$,

***$P<0.001$.

# Adjusted for age, gender, race, PIR, physical activity, sedentary time, and maternal smoking during pregnancy.

Note: We used g/1000kcal as the units of Ca, P, Mg, Na and K to expanded the value.

obesity in youngsters. Cai et al. [24] indicated that high dietary K intake could not reduce the risk of obesity, while serum K and urinary Na-to-K ratio was associated with obesity. In our study, the average dietary consumptions of Ca, Fe, Zn and Cu were significantly lower in children with obesity than those without obesity, and consumptions of Na and Se were higher in children with obesity. The possible reason for these inconsistent results, we speculated, may be that children with obesity are more likely to control the quantity of dietary minerals intakes, such as Cu and Ca, following medical advices. Moreover, the specific mechanisms on associations between dietary intakes of minerals are needed to be further clarified.

Obesity is characterized by a low-grade systemic chronic inflammatory state [25]. Constant macrophage infiltrate into adipose tissue and the local pro-inflammatory cytokines change in

**Table 5. Association between dietary minerals intake and BMI in age subgroups.**

| Dietary minerals intake | 2–5 years old β (95% CI) | 6–11 years old β (95% CI) | 12–17 years old β (95% CI) |
|---|---|---|---|
| Ca (mg/1000kcal) | | | |
| ≤391.56 | Ref | Ref | Ref |
| 391.56 to 544.69 | -0.05(-0.32, 0.22) | -0.06(-0.45, 0.32) | 0.12(-0.62, 0.85) |
| 544.69 to 721.81 | -0.06(-0.31, 0.19) | -0.04(-0.48, 0.40) | 0.46(-0.36, 1.28) |
| >721.81 | 0.07(-0.18, 0.32) | -0.19(-0.85, 0.47) | 0.78(-0.13, 1.69) |
| P (mg/1000kcal) | | | |
| ≤549.25 | Ref | Ref | Ref |
| 549.25 to 661.63 | -0.08(-0.37, 0.22) | -0.35(-0.77, 0.06) | -0.61(-1.36, 0.15) |
| 661.63 to 786.65 | -0.10(-0.36, 0.15) | -0.20(-0.77, 0.34) | 0.18(-0.69, 1.05) |
| >786.65 | 0.02(-0.28, 0.32) | -0.62(-1.30, 0.07) | 0.44(-0.50, 1.37) |
| Mg (mg/1000kcal) | | | |
| ≤100.41 | Ref | Ref | Ref |
| 100.41 to 122.18 | -0.06(-0.34, 0.23) | -0.07(-0.50, 0.37) | -0.18(-1.15, 0.80) |
| 122.18 to 146.32 | -0.12(-0.45, 0.21) | 0.08(-0.49, 0.65) | 0.59(-0.29, 1.47) |
| >146.32 | 0.02(-0.34, 0.38) | 0.31(-0.27, 0.89) | 0.10(-0.71, 0.92) |
| Fe (mg/1000kcal) | | | |
| ≤5.39 | Ref | Ref | Ref |
| 5.39 to 6.85 | -0.12(-0.37, 0.15) | 0.14(-0.37, 0.64) | -0.63(-1.44, 0.18) |
| 6.85 to 9.33 | -0.27(-0.49, -0.06) * | -0.02(-0.39, 0.36) | -1.09(-1.70, -0.48) *** |
| >9.33 | -0.29(-0.56, -0.03) * | -0.51(-1.04, 0.02) | -1.31(-2.11, -0.52) ** |
| Zn (mg/1000kcal) | | | |
| ≤4.01 | Ref | Ref | Ref |
| 4.01 to 5.20 | 0.23(-0.03, 0.48) | -0.39(-0.93, 0.15) | -0.22(-1.07, 0.63) |
| 5.20 to 7.08 | 0.12(-0.14, 0.38) | -0.32(-0.91, 0.26) | -0.42(-1.34, 0.51) |
| >7.08 | -0.06(-0.28, 0.16) | -1.04(-1.65, -0.43) ** | -0.44(-1.37, 0.48) |
| Cu (mg/1000kcal) | | | |
| ≤0.41 | Ref | Ref | Ref |
| 0.41 to 0.50 | 0.06(-0.16, 0.28) | 0.49(0.04, 0.94) * | 0.00(-0.87, 0.88) |
| 0.50 to 0.64 | 0.00(-0.20, 0.20) | 0.61(0.09, 1.13) * | -0.12(-0.92, 0.67) |
| >0.64 | -0.10(-0.39, 0.19) | 0.07(-0.41, 0.56) | -0.25(-1.06, 0.56) |
| Na (mg/1000kcal) | | | |
| ≤1282.86 | Ref | Ref | Ref |
| 1282.86 to 1541.84 | 0.01(-0.23, 0.24) | 0.31(-0.12, 0.75) | 0.26(-0.72, 1.24) |
| 1541.84 to 1819.50 | 0.04(-0.21, 0.30) | -0.24(-0.75, 0.27) | 0.04(-0.65, 0.74) |
| >1819.50 | 0.12(-0.16, 0.41) | 0.65(0.05, 1.25) * | 0.84(0.01, 1.66) * |
| K (mg/1000kcal) | | | |
| ≤916.70 | Ref | Ref | Ref |
| 916.70 to 1141.73 | 0.06(-0.25, 0.38) | 0.03(-0.39, 0.46) | 0.13(-0.64, 0.89) |
| 1141.73 to 1407.27 | -0.11(-0.36, 0.14) | 0.04(-0.49, 0.57) | 0.74(-0.03, 1.51) |
| >1407.27 | 0.07(-0.18, 0.32) | 0.15(-0.49, 0.79) | 0.54(-0.37, 1.45) |
| Se (mcg/1000kcal) | | | |
| ≤37.75 | Ref | Ref | Ref |
| 37.75 to 48.05 | -0.07(-0.31, 0.18) | -0.11(-0.58, 0.36) | -0.36(-1.37, 0.64) |
| 48.05 to 58.52 | -0.11(-0.49, 0.26) | -0.29(-0.79, 0.21) | -0.34(-1.49, 0.82) |

(*Continued*)

**Table 5.** (Continued)

| Dietary minerals intake | 2–5 years old β (95% CI) | 6–11 years old β (95% CI) | 12–17 years old β (95% CI) |
|---|---|---|---|
| >58.52 | -0.08(-0.40, 0.25) | 0.12(-0.52, 0.76) | -0.01(-1.05, 1.03) |

BMI: Body mass index, CI: Confidence interval, Ca: Calcium, Ref: Reference, P: Phosphorus, Mg: Magnesium, Fe: Iron, Zn: Zinc, Cu: Copper, Na: Sodium, K: Potassium and Se: Selenium.

*$P<0.05$,

**$P<0.01$,

***$P<0.001$.

Adjusted for gender, race, PIR, physical activity, sedentary time, and maternal smoking during pregnancy.

Note: We used g/1000kcal as the units of Ca, P, Mg, Na and K to expanded the value.

obesity may lead to impaired erythropoietin production and altered response of erythroid precursors, which has been a recognized mechanism of anemia associated with chronic diseases [4,26]. Studies have reported that lower serum Fe concentrations and transferrin saturation were found in obese/overweight individuals, and a negative correlation between transferrin saturation and adiposity was also found in school children with obesity who aged between 9 and 13 years old [27,28]. In the current study, we are unavailable to obtain the anemia-related indexes such as serum Fe and transferrin saturation between children with obesity and without obesity. Basing on a previous study, we speculated that lack of dietary Fe may disturb the hemoproteins or other non-heme proteins metabolism, and affect the formation of toxic oxygen free radicals that further influence the development of obesity [29]. Analogously, Zn is an efficient antioxidant and plays a major role in some protein productions such as insulin action and insulin receptor tyrosine kinase activity [30]. We found that higher dietary Zn intake was associated with obesity and lower BMI in children. The underlying biological mechanism could be the gene expression of Zn-a2-glycoprotein (ZAG) is lower in subcutaneous and visceral adipose tissue and livers of obese individuals, which may play an important role in the development of obesity [31]. Up to now, the potential mechanisms on a direct association between dietary Na intake and obesity are still unclear. Epidemiologic studies suggested that salt intake was associated with leptin concentrations, percentage of body fat, and adipose tissue [32,33]. High-salt diets may contribute to the progression of obesity by increasing fasting ghrelin, which regulates appetite, glucose homeostasis, and fat deposition [34]. Whether these mechanisms are appropriate for the explanation on our findings are needed further basic research to verify. In addition, Cu is a redox-active metal, and elevated Cu in circulation may contribute to exacerbate oxidative stress by generating free radical species, thereby inducing lipid peroxidation. In this context, we suggested that the abnormal lipid profiles associated with Cu overload may lead to the obesity related systemic inflammation and oxidative stress [16,35].

The associations between higher dietary intake of Zn and lower odds of childhood obesity, as well as between higher dietary intakes of Mg and Na and higher odds of childhood obesity were also found in children aged 6–11 years old. The negative relationship between dietary Fe intake and BMI was found in children who aged 2–5 years old, and 12–17 years old. The negative relationship between dietary Zn intake and BMI was only found in those who aged 6–11 years old. Also, positive association between dietary Na intake and BMI was found in children who aged from 6 to 17 years old. For children with different age, it is necessary to develop individualization recommendations of minerals intake in obesity. After 2 years old, not only dietary patterns (including minerals consumption), but also sugar-sweetened beverages

consumption, eating behavior, meal frequency and composition, portion size and physical activity, and sedentary behavior were the influencing factors of childhood obesity [6,36]. To prevent and manage children with overweight/obesity is a long-term, multi-stage process. A balanced diet with fiber-rich foods including whole grains, lentils, nuts, fruits and vegetables benefits to improve and prevent obesity [37]. Combining our findings, children aged 2–5 years old are recommended to get plenty of iron-containing fruits and vegetables, whereas children aged 6–11 years old are recommended for foods are rich in Zn. Snacks with high Na content should be strictly controlled at all ages during childhood, and a regular meal pattern along with healthy habits were also needed [38]. In addition to dietary, children should optimally have 60 mins of moderate to vigorous physical activity, at least 5 days per week, to decrease the risk of developing obesity [39]. Besides, school-age children and adolescents should sleep for 8 to 11 hours per day and in quiet surroundings, due to insufficient sleep can affect dietary intake and metabolism, which may lead to obesity [40].

This study population was from the NHANES database, so that the sample size was large and were the representative population in the United States. We explored the association between dietary intakes of nine common minerals and childhood obesity and BMI, which may provide some references for further studies exploring the causal associations, and may further help the prevention and management of childhood obesity. However, there are some limitations in this study. This is a retrospective study so that no causal association could be concluded. Information of dietary intake were collected using the 24-hour recall, which can cause the recalling bias, and at the same time, could not reflect the long-term dietary habits. Therefore, further prospective cohort studies focusing on the long-term effects of dietary minerals intake on childhood obesity are still needed.

## Conclusion

Dietary Fe and Zn intakes may benefit to reduce the odds of childhood obesity, whereas the dietary Cu and Na consumptions should be controlled suitably. In addition, it is necessary to develop individualization recommendations of minerals intake for children with high-risk of obesity in different age in the future.

## Supporting information

**S1 Checklist. STROBE statement—Checklist of items that should be included in reports of observational studies.**
(DOCX)

## Author Contributions

**Conceptualization:** Lu Wang.

**Data curation:** Wei Liu, Sitong Bi, Li Zhou, Lihua Li.

**Formal analysis:** Wei Liu, Sitong Bi, Li Zhou, Lihua Li.

**Investigation:** Wei Liu, Sitong Bi, Li Zhou, Lihua Li.

**Methodology:** Wei Liu, Sitong Bi, Li Zhou, Lihua Li.

**Writing – original draft:** Lu Wang.

**Writing – review & editing:** Lu Wang.

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
