## [Decision Letter · Decision Letter 0]

23 Jun 2023

PONE-D-23-10754Association between minerals intake and the risk of childhood obesity: a cross-sectional study from the National Health and Nutrition Examination Survey 2007-2014PLOS ONE

Dear Dr. Wang,

Thank you for submitting your manuscript to PLOS ONE. After careful consideration, we feel that it has merit but does not fully meet PLOS ONE’s publication criteria as it currently stands. Therefore, we invite you to submit a revised version of the manuscript that addresses the points raised during the review process.

We look forward to receiving your revised manuscript.

Kind regards,

Zhe He, PhD

Academic Editor

PLOS ONE

Journal Requirements:

2. PLOS requires an ORCID iD for the corresponding author in Editorial Manager on papers submitted after December 6th, 2016. Please ensure that you have an ORCID iD and that it is validated in Editorial Manager. To do this, go to ‘Update my Information’ (in the upper left-hand corner of the main menu), and click on the Fetch/Validate link next to the ORCID field. This will take you to the ORCID site and allow you to create a new iD or authenticate a pre-existing iD in Editorial Manager. Please see the following video for instructions on linking an ORCID iD to your Editorial Manager account: https://www.youtube.com/watch?v=_xcclfuvtxQ.

Additional Editor Comments:

Both reviewers have raised a number of issues. The analysis needs to be re-done by carefully choosing variables and removing duplicated variables. English has to be improved for better readability.

Reviewers' comments:

Reviewer's Responses to Questions

**Comments to the Author**

1. Is the manuscript technically sound, and do the data support the conclusions?

Reviewer #1: Partly

Reviewer #2: Partly

2. Has the statistical analysis been performed appropriately and rigorously? 

Reviewer #1: Yes

Reviewer #2: No

3. Have the authors made all data underlying the findings in their manuscript fully available?

Reviewer #1: Yes

Reviewer #2: Yes

4. Is the manuscript presented in an intelligible fashion and written in standard English?

Reviewer #1: No

Reviewer #2: No

5. Review Comments to the Author

Reviewer #1: Line 49 - Is it "in 2025" or "by 2025"?

Line 60 - Should be play instead of played.

Line 63-66 - No mention/discussion of why sodium intake is high.

Line 73 - What was the reasoning behind picking the minerals that you did and excluding others?

Line 90-93 - You need to be clearer on exclusions from the study. The figure includes BMI not height and weight and mentions PIR, which is not noted in lines 90-93.

Lines 99-116 - This section needs to be a better description of your methods.

Line 119 - Should be children's BMI instead of children BMI

Line 158-160 - Explain more about the adjustments that were made.

Line 173 - Should be with obesity and without instead of have and not have.

Line 174 - Random question mark in text.

Line 175 - Should be children with obesity not obese children.

Line 176 - Should be while those without obesity included a higher percentage of females.

Line 176 - Should be children without obesity instead of obese children. (This error is continued throughout)

Line 177 - Should be had an ideal physical activity level, while in children with obesity this number was …..

Line 191 - Should this be risk of childhood obesity or incidence of childhood obesity (This use of risk is repeated throughout)

Line 272/344 - Should be risk/incidence of obesity and higher BMI.

Line 272 - High/higher/normal level/advised level of dietary FE. How do your findings coincide with the recommendations for intake of these minerals for each age group.

Line 277 - Should be in different age groups not with different age (Repeated several times)

Line 278-292 - You need to better relate your study in context to these other studies. How is yours different? Did you come to the same conclusions or different conclusions?

Line 294-317 - Organize this paragraph better so it does not seem like you are merely stating facts. This is the discussion section so there needs to be a discussion between this study and the findings and the facts that you are providing through references.

Line 330 - Are you stating that mineral consumption plays a lesser role after the age of 2?

Line 333 - Start a new paragraph and relate this information to your study.

Line 345 - I did not get this information from your paper. There was no dietetic reference presented and nothing related to management of obesity.

Line 353 - Should be intakes may benefit not be benefit.

Line 381 - This citation is not correct.

Reviewer #2: In this paper, the authors investigated the association between minerals and the risk of obesity and body mass index (BMI) in children with different age based on NHANES data. I have some major concerns:

1. Line 146, the authors mentioned “Non-normal distribution data were described by median and quartiles [M (Q1, Q3)] and Mann-Whitney U rank test for the comparation.” Instead of testing the normality, I would suggest the authors show key statistics for all numerical variables, including mean, standard deviation, median, q1 and q3. They are important and helpful for readers to understand the distribution.

2. In the section of “Association between dietary minerals intake and the risk of childhood obesity”, it seems one mineral was included in the model as two types of variables: numerical and categorical types. For example, both of numerical Ca and categorical Ca grouped as four levels were included. It is confused. Such model setting may also cause multicollinearity. I would suggest you focus on either one. You can investigate the association between different levels of minerals and risk of childhood obesity, or whether the amount of intake is associated with risk of obesity. Other models also had such issue.

3. In the sections of “Association between dietary minerals intake and BMI in children” and “Association between dietary minerals intake and BMI in age subgroups”, I don’t understand why you use beta not OR? Beta is logistic regression coefficient and not straightforward to interpret compared with OR.

4 Please check your grammar carefully.

5 Please improve the scientific writing and make the manuscript more readable. I just listed a part which should be refined.

line 90 “The participants’ included criteria”, should be “The inclusion criteria …

line 101 “24-h dietary recall interviews” Does 24-h means 24-hour? Please make the abbreviation more clear.

line 118 Pay attention to singular and plural: “Outcome variable” misses “s”.

l144 “Normality of quantitative data was test by Kolmogorov-Smirnov test.”

l145 “The normal distribution data were described using mean ± standard error (mean ± SE) and that t test for group comparation”

l155 “Weighted univariate logistic regression analyses were used to screen the covariates that affect the BMI.”

l170 “The dietary intake levels of Ca (578.39 mg/1000kcal vs. 559.39 mg/1000kcal)” I would suggest to be “(578.39 vs. 559.39 mg/1000kcal)”

l173 “The characteristics of children have obesity and not have obesity with and without obesity? are showed in Table 1.”

l176 “Most obese children were male [1048 (53.49%)] while that non-obese were female [4150 (50.12%)].”

l183 “In addition, race, PIR, sedentary time, maternal smoking during pregnancy, cotinine, intakes of protein and carbohydrate were statistical differences between the two groups as well (all P<0.05).”

l205 table2 don’t use one mark(*) to indicate multiple footnotes.

l338 what is “at least 5 d per week” ?

l339 what is “8 to 11 h a day”?

page 41 figure 1 miss blank: “from 2007 to2014”

6. PLOS authors have the option to publish the peer review history of their article (what does this mean?). If published, this will include your full peer review and any attached files.

Reviewer #1: No

Reviewer #2: No

---

## [Author Response · Author response to Decision Letter 0]

3 Aug 2023

Responses to Reviewers

Dear Editor and reviewers,

We highly appreciate and thank the Editors and the reviewers for your time and valuable comments on this manuscript. We have carefully studied the comments and suggestions and revised our paper accordingly. The following are our point-by-point responses to the comments. We hope that the revisions are acceptable and that our responses adequately address the comments. Thank you for your consideration.

Responses to Reviewer #1’s comments

Line 49 - Is it "in 2025" or "by 2025"?

Response: Thank you for your comment. We have revised “in 2025” into “by 2025.”

Line 60 - Should be play instead of played.

Response: Thank you for your comment. We have revised “played” into “play.”

Line 63-66 - No mention/discussion of why sodium intake is high.

Response: We revised the description for association between dietary sodium intake and the risk of obesity. These sentences were aim to introduce the effects of minerals on obesity and the potential mechanism.

Line 73 - What was the reasoning behind picking the minerals that you did and excluding others?

Response: Thank you for your comment. This cross-sectional study was based on the NHANES database, and in which, all the records for dietary minerals included Ca, P, Mg, Fe, Zn, Cu, Na, K and Se. We could not obtain other minerals intake status, rather than excluding others.

Line 90-93 - You need to be clearer on exclusions from the study. The figure includes BMI not height and weight and mentions PIR, which is not noted in lines 90-93.

Response: Thank you for your comment. We have revised the description of inclusions and exclusions for the study population.

Lines 99-116 - This section needs to be a better description of your methods.

Response: Thank you for your comment. We have revised this section to describe the methods better.

Line 119 - Should be children's BMI instead of children BMI

Response: Thank you for your comment. We have changed the “children BMI” into “children’s BMI.”

Line 158-160 - Explain more about the adjustments that were made.

Response: Thank you for your comment. We have added some explanations for the adjustments.

Line 173 - Should be with obesity and without instead of have and not have.

Response: Thank you for your comment. We have revised the “have and not have” into “with and without.”

Line 174 - Random question mark in text.

Response: We are so sorry for this clerical error, and the question mark has been deleted.

Line 175 - Should be children with obesity not obese children.

Response: Thank you for your comment. We have revised the “obese children” into “children with obesity.”

Line 176 - Should be while those without obesity included a higher percentage of females.

Response: Thank you for your comment. We have revised the sentence into “hile those without obesity included a higher percentage of females.”

Line 176 - Should be children without obesity instead of obese children. (This error is continued throughout)

Response: Thank you for your comment. We have corrected this error in the whole manuscript.

Line 177 - Should be had an ideal physical activity level, while in children with obesity this number was …

Response: Thank you for your comment. We have changed this sentence into “had an ideal physical activity level, while in children with obesity this number was 741 (31.90%).”

Line 191 - Should this be risk of childhood obesity or incidence of childhood obesity (This use of risk is repeated throughout)

Response: Thank you for your comment. Due to this was a cross-sectional study which can not explore the risk of incidence of childhood obesity, and we have changed the descriptions for study results using “odds” instead of “risk”. We explored the association between dietary minerals intake and childhood obesity, and the “childhood obesity” in the study was only reflected the current status of whether children with obesity or without.

Line 272/344 - Should be risk/incidence of obesity and higher BMI.

Response: Thank you for your comment. We have changed this sentence into “This study explored the association between dietary intakes of nine common minerals and obesity and higher BMI.”

Line 272 - High/higher/normal level/advised level of dietary FE. How do your findings coincide with the recommendations for intake of these minerals for each age group.

Response: Thank you for your comment. This study aimed to explore associations between nine common minerals and childhood obesity and BMI. We divided each mineral intakes into four levels according to their quartiles basing on the status of consumption of minerals among the study population. Due to it was a cross-sectional study, we could only assess the current mineral intake levels in children aged 2-17 years old in NHANES. The high level of dietary mineral intakes represented the highest quartile of the consumption level comparing to the lowest in the study population. We think your comment about whether the dietary intake levels of mineral can coincide with their recommendations for each age group is valuable, and further study could focus on this to provide some reference for the personalized dietary mineral supplement for different ages.

Line 277 - Should be in different age groups not with different age (Repeated several times)

Response: Thank you for your comment. We have revised this error in the whole manuscript.

Line 278-292 - You need to better relate your study in context to these other studies. How is yours different? Did you come to the same conclusions or different conclusions?

Response: Thank you for your comment. We have revised this paragraph to context our study to other studies, and compared ours with others’.

Line 294-317 - Organize this paragraph better so it does not seem like you are merely stating facts. This is the discussion section so there needs to be a discussion between this study and the findings and the facts that you are providing through references.

Response: Thank you for your comment. We have organized this paragraph better to discuss between the findings of this study and the facts providing through references.

Line 330 - Are you stating that mineral consumption plays a lesser role after the age of 2?

Response: Thank you for your comment. In this paragraph, we discussed the results of subgroup analyses. The study population was children who aged from 2 to 17 years old, and we divided them into three age subgroups to explore the associations between minerals consumption and childhood obesity and BMI, with the aim of providing some reference for individualization recommendations of minerals intake in children have the risk of obesity.

Line 333 - Start a new paragraph and relate this information to your study.

Response: Thank you for your comment. We have started a new paragraph and related this information to our study.

Line 345 - I did not get this information from your paper. There was no dietetic reference presented and nothing related to management of obesity.

Response: Thank you for your comment. This study explored the associations between minerals intake and childhood obesity and children’s BMI. We found some potential relationships including positive and negative, and hope it may provide some reference for the further studies which aim to analyze whether there is a causal link between them, and in turn can provide information for the dietary management in children with obesity.

Line 353 - Should be intakes may benefit not be benefit.

Response: Thank you for your comment. We have revised the “may be benefit” into “may benefit.”

Line 381 - This citation is not correct.

Response: Thank you for your comment. We have revised this citation.

Responses to Reviewer #2’s comments

In this paper, the authors investigated the association between minerals and the risk of obesity and body mass index (BMI) in children with different age based on NHANES data. I have some major concerns:

1. Line 146, the authors mentioned “non-normal distribution data were described by median and quartiles [M (Q1, Q3)] and Mann-Whitney U rank test for the comparation.” Instead of testing the normality, I would suggest the authors show key statistics for all numerical variables, including mean, standard deviation, median, q1 and q3. They are important and helpful for readers to understand the distribution.

Response: Thank you for your comment. We have revised the description of statistics for all numerical variables to help for readers to understand the distribution.

2. In the section of “Association between dietary minerals intake and the risk of childhood obesity”, it seems one mineral was included in the model as two types of variables: numerical and categorical types. For example, both of numerical Ca and categorical Ca grouped as four levels were included. It is confused. Such model setting may also cause multicollinearity. I would suggest you focus on either one. You can investigate the association between different levels of minerals and risk of childhood obesity, or whether the amount of intake is associated with risk of obesity. Other models also had such issue.

Response: Thank you for your comment. This study was aim to explore the association between dietary minerals intake and childhood obesity, in which the dietary minerals intake was included as both numerical and categorical types. The numerical type in model can reflected the change of odds of childhood obesity along with the change of dietary minerals intake per unit, while the categorical type in model can reflected the odds of childhood obesity at different levels of dietary minerals intake. We have deleted the numerical type of dietary minerals intake in models to make it not confused for reader and avoid the multicollinearity.

3. In the sections of “Association between dietary minerals intake and BMI in children” and “Association between dietary minerals intake and BMI in age subgroups”, I don’t understand why you use beta not OR? Beta is logistic regression coefficient and not straightforward to interpret compared with OR.

Response: Thank you for your comment. We are so sorry that we have not clarify the analytical methods for exploration of the association between dietary minerals intake and childhood obesity and BMI respectively in the Methods section. Due to the “BMI” was a continuous variable, we used the univariate and multivariate linear regression analyses for the statistical analyses. The “β” was the estimated value in linear regression analyses, which >0 or <0 can reflected the association between dietary minerals intake and BMI was positively or negatively.

4 Please check your grammar carefully.

Response: Thank you for your comment. We have checked the grammar and revised errors in the manuscript.

5 Please improve the scientific writing and make the manuscript more readable. I just listed a part which should be refined.

Response: Thank you for your comment. We have revised each point you mentioned and improved the scientific writing of the manuscript.

line 90 “The participants’ included criteria”, should be “The inclusion criteria …

Response: Thank you for your comment. We have revised this sentence.

line 101 “24-h dietary recall interviews” Does 24-h means 24-hour? Please make the abbreviation clearer.

Response: We have changed the “24-h” into “24-hour.”

line 118 Pay attention to singular and plural: “Outcome variable” misses “s”.

Response: We have changed the “Outcome variable” into “Outcome variables.”

l144 “Normality of quantitative data was test by Kolmogorov-Smirnov test.”

Response: We have revised this sentence.

l145 “The normal distribution data were described using mean ± standard error (mean ± SE) and that t test for group comparation”

Response: We have revised this sentence.

l155 “Weighted univariate logistic regression analyses were used to screen the covariates that affect the BMI.”

Response: We have revised this sentence.

l170 “The dietary intake levels of Ca (578.39 mg/1000kcal vs. 559.39 mg/1000kcal)” I would suggest to be “(578.39 vs. 559.39 mg/1000kcal)”

Response: We have revised these sentences.

l173 “The characteristics of children have obesity and not have obesity with and without obesity? are showed in Table 1.”

Response: We have revised this sentence.

l176 “Most obese children were male [1048 (53.49%)] while that non-obese were female [4150 (50.12%)].”

Response: We have revised this sentence.

l183 “In addition, race, PIR, sedentary time, maternal smoking during pregnancy, cotinine, intakes of protein and carbohydrate were statistical differences between the two groups as well (all P<0.05).”

Response: We have revised this sentence.

l205 table2 don’t use one mark (*) to indicate multiple footnotes.

Response: We have used different marks to indicate multiple footnotes.

l338 what is “at least 5 d per week”?

Response: We have revised “at least 5 d per week” into “at least 5 days per week.”

l339 what is “8 to 11 h a day”?

Response: We have revised “8 to 11 h a day” into “8 to 11 hours a day.”

page 41 figure 1 miss blank: “from 2007 to2014” 

Response: We have added the blank: “from 2007 to2014.”

---

## [Decision Letter · Decision Letter 1]

8 Nov 2023

PONE-D-23-10754R1Association between minerals intake and childhood obesity: a cross-sectional study of the NHANES database in 2007-2014PLOS ONE

Dear Dr. Wang,

Thank you for submitting your manuscript to PLOS ONE. After careful consideration, we feel that it has merit but does not fully meet PLOS ONE’s publication criteria as it currently stands. Therefore, we invite you to submit a revised version of the manuscript that addresses the points raised during the review process.

We look forward to receiving your revised manuscript.

Kind regards,

Zhe He, PhD

Academic Editor

PLOS ONE

Journal Requirements:

Additional Editor Comments :

The reviewers still have some minor remarks to be addressed.

Reviewers' comments:

Reviewer's Responses to Questions

**Comments to the Author**

1. If the authors have adequately addressed your comments raised in a previous round of review and you feel that this manuscript is now acceptable for publication, you may indicate that here to bypass the “Comments to the Author” section, enter your conflict of interest statement in the “Confidential to Editor” section, and submit your "Accept" recommendation.

Reviewer #1: (No Response)

Reviewer #2: (No Response)

2. Is the manuscript technically sound, and do the data support the conclusions?

Reviewer #1: Yes

Reviewer #2: Yes

3. Has the statistical analysis been performed appropriately and rigorously? 

Reviewer #1: (No Response)

Reviewer #2: Yes

4. Have the authors made all data underlying the findings in their manuscript fully available?

Reviewer #1: Yes

Reviewer #2: Yes

5. Is the manuscript presented in an intelligible fashion and written in standard English?

Reviewer #1: No

Reviewer #2: No

6. Review Comments to the Author

Reviewer #1: This manuscript is still lacking in areas related to coherency and grammar. From the Abstract to the Conclusion there are still areas that need to be written better. It appears that the listed changes were made but that the overall manuscript was not fully edited. I also feel that the discussion section is not as approachable as it needs to be. Readers need to be able to fully understand both the results and the contribution after reading the discussion.

Reviewer #2: 1 In section of “Statistical analysis”: you should mention clearly which reference group in the dependent variable is

2 Table 4 and table 5:

Round numbers to 2 decimal places throughout the manuscript to keep consistent.

3 The authors should improve the scientific writing and check text carefully before submission:

line 120 formula: weight/Height2 (kg/m2): should be “height”

line 124 that ≥95th percentile indicates obesity: should be “that of…”

line 174 obesity? are showed in Table 1: display incorrectly

line 176 If you decided to use comma for numbers with 4 or more digits, please be consistent throughout the manuscript

7. PLOS authors have the option to publish the peer review history of their article (what does this mean?). If published, this will include your full peer review and any attached files.

Reviewer #1: No

Reviewer #2: No

---

## [Author Response · Author response to Decision Letter 1]

9 Nov 2023

Responses to Reviewer #1’s comments

This manuscript is still lacking in areas related to coherency and grammar. From the Abstract to the Conclusion there are still areas that need to be written better. It appears that the listed changes were made but that the overall manuscript was not fully edited. I also feel that the discussion section is not as approachable as it needs to be. Readers need to be able to fully understand both the results and the contribution after reading the discussion.

Response: Thank you for your valuable comments. We have reviewed and revised the manuscript to make it more coherent and easier to understand. We also invited a native English-speaker to help us revise, and we hope the revised manuscript could achieve your standards.

Responses to Reviewer #2’s comments

1 In section of “Statistical analysis”: you should mention clearly which reference group in the dependent variable is

Response: We have added the description on reference group.

2 Table 4 and table 5:

Round numbers to 2 decimal places throughout the manuscript to keep consistent.

Response: Thank you for your valuable comment. We have revised the tables to keep the numbers to 2 decimal places consistently.

3 The authors should improve the scientific writing and check text carefully before submission:

line 120 formula: weight/Height2 (kg/m2): should be “height”

line 124 that ≥95th percentile indicates obesity: should be “that of…”

line 174 obesity? are showed in Table 1: display incorrectly

line 176 If you decided to use comma for numbers with 4 or more digits, please be consistent throughout the manuscript

Response: Thank you for your valuable comments. The above mistakes have been revised, and we have checked and edited the manuscript.

---

## [Editor Report · Decision Letter 2]

29 Nov 2023

Association between minerals intake and childhood obesity: a cross-sectional study of the NHANES database in 2007-2014

PONE-D-23-10754R2

Dear Dr. Wang,

We’re pleased to inform you that your manuscript has been judged scientifically suitable for publication and will be formally accepted for publication once it meets all outstanding technical requirements.

Kind regards,

Zhe He, PhD

Academic Editor

PLOS ONE

Additional Editor Comments (optional):

The authors have addressed all the issues raised by the reviewers.

---

## [Editor Report · Acceptance letter]

4 Dec 2023

PONE-D-23-10754R2 

Association between minerals intake and childhood obesity: a cross-sectional study of the NHANES database in 2007-2014 

Dear Dr. Wang:

I'm pleased to inform you that your manuscript has been deemed suitable for publication in PLOS ONE. Congratulations! Your manuscript is now with our production department. 

Kind regards, 

on behalf of

Dr. Zhe He 

Academic Editor

PLOS ONE